# GENERALIZED CONVERGENCE ANALYSIS OF TSETLIN AUTOMATON BASED ALGORITHMS: A PROBABILISTIC APPROACH TO CONCEPT LEARNING

## ABSTRACT

Tsetlin Machines (TMs) have garnered increasing interest for their ability to learn concepts via propositional formulas and their proven efficiency across various application domains. Despite this, the convergence proof for the TMs, particularly for the AND operator (*conjunction* of literals), in the generalized case (inputs greater than two bits) remains an open problem. This paper aims to fill this gap by presenting a comprehensive convergence analysis of Tsetlin automaton-based Machine Learning algorithms. We introduce a novel framework, referred to as Probabilistic Concept Learning (PCL), which simplifies the TM structure while incorporating dedicated feedback mechanisms and dedicated inclusion/exclusion probabilities for literals. Given $n$ features, PCL aims to learn a set of conjunction clauses $C_i$ each associated with a distinct inclusion probability $p_i$. Most importantly, we establish a theoretical proof confirming that, for any clause $C_k$, PCL converges to a conjunction of literals when $0.5 < p_k < 1$. This result serves as a stepping stone for future research on the convergence properties of Tsetlin automaton-based learning algorithms. Our findings not only contribute to the theoretical understanding of Tsetlin automaton-based learning algorithms but also have implications for their practical application, potentially leading to more robust and interpretable machine learning models.

## 1 INTRODUCTION

Concept Learning, a mechanism to infer Boolean functions from examples, has its foundations in classical machine learning Valiant (1984); Angluin (1988); Mitchell (1997). A modern incarnation, the Tsetlin Machine (TM) Granmo (2018), utilizes Tsetlin Automata (TAs) Tsetlin (1961) to generate Boolean expressions as conjunctive clauses. Contrary to the opaqueness of deep neural networks, TMs stand out for their inherent interpretability rooted in disjunctive normal form Valiant (1984). Recent extensions to the basic TM include architectures for convolution Granmo et al. (2019), regression Abeyrathna et al. (2020), and other diverse variants Seraj et al. (2022); Sharma et al. (2023); Abeyrathna et al. (2021; 2023). These advances have found relevance in areas like sentiment analysis Yadav et al. (2021) and novelty detection Bhattarai et al. (2022).

Proving the convergence of a machine learning model is crucial as it guarantees the model's reliability and stability, ensuring that it reaches a consistent solution Shalev-Shwartz et al. (2010); Berkenkamp et al. (2017). It also aids in the development and evaluation of algorithms, providing a theoretical benchmark for performance and understanding the model's limitations. Convergence analysis of TMs reveals proven behavior for 1-bit Zhang et al. (2022) and 2-bit cases Jiao et al. (2021; 2023), encompassing the AND, OR, and XOR operators. However, general convergence, especially with more input bits, poses significant challenges. The crux of the issue stems from the clause-based interdependence of literals in the learning mechanism of TMs. Essentially, the feedback to a literal is influenced by other literals in the same clause. This interdependency, combined with vast potential combinations of literals, compounds the difficulty in a general proof for TMs.

Addressing this, our work introduces Probabilistic inclusion of literals for Concept Learning (PCL), an innovative TM variant. PCL's design allows literals to be updated and analyzed independently, contrasting starkly with standard TM behavior. This is achieved by tweaking feedback tables to

Figure 1: A two-action Tsetlin automaton with $2N$ states Jiao et al. (2023).

exclude clause values during training and omitting the inaction transition. Additionally, PCL employs dedicated inclusion probabilities for clauses to diversify the learned patterns. Most importantly, we provide evidence that PCL clauses, under certain preconditions, can converge to any intended conjunction of literals. Our assertions are bolstered by experimental results to confirm the theoretical finding. Finally, it is important to note that our proof on the convergence of PCL does not imply convergence of the original Tsetlin Machine, but it lays a robust foundation and outlines a clear roadmap for potential proofs concerning learning algorithms that are based on Tsetlin Automaton.

## 2 TSETLIN MACHINE

**Structure:** A Tsetlin Machine (TM) processes a boolean feature vector $\mathbf{x} = [x_1, \ldots, x_n] \in \{0, 1\}^n$ to assign a class $\hat{y} \in \{0, 1\}$. The vector defines the literal set $L = \{x_1, \ldots, x_n, \neg x_1, \ldots, \neg x_n\}$. TM uses subsets $L_j \subseteq L$ to generate patterns, forming conjunctive clauses:

$$C_j(\mathbf{x}) = \bigwedge_{l_k \in L_j} l_k. \tag{1}$$

A clause $C_j(\mathbf{x}) = x_1 \wedge \neg x_2$, for instance, evaluates to 1 when $x_1 = 1$ and $x_2 = 0$.

Each clause $C_j$ uses a Tsetlin Automata (TA) for every literal $l_k$. This TA determines if $l_k$ is *Excluded* or *Included* in $C_j$. Figure 1 shows the states of a TA with two actions. Each TA chooses one of two actions, i.e., either *Includes* or *Excludes* its associated literal. When the TA is in any state between 1 to $N$, the action *Exclude* is selected. Likewise, the action becomes *Include* when the TA is in any state between $N + 1$ to $2N$. The transitions among the states are triggered by a reward or a penalty that the TA receives from the environment.

With the $u$ clauses and $2n$ literals, there are $u \times 2n$ TAs. Their states are organized in the matrix $A = [a_k^j] \in \{1, \ldots, 2N\}^{u \times 2n}$. The function $g(\cdot)$ maps the state $a_k^j$ to actions *Exclude* (for states up to $N$) and *Include* (for states beyond $N$).

Connecting the TA states to clauses, we get:

$$C_j(\mathbf{x}) = \bigwedge_{k=1}^{2n} \left[ g(a_k^j) \Rightarrow l_k \right]. \tag{2}$$

The *imply* operator governs the *Exclude/Include* action, being 1 if excluded, and the literal's truth value if included.

**Classification:** Classification uses a majority vote. Odd-numbered clauses vote for $\hat{y} = 0$ and even-numbered ones for $\hat{y} = 1$. The formula is:

$$\hat{y} = 0 \leq \sum_{j=1,3,\ldots}^{u-1} \bigwedge_{k=1}^{2n} \left[ g(a_k^j) \Rightarrow l_k \right] - \sum_{j=2,4,\ldots}^{u} \bigwedge_{k=1}^{2n} \left[ g(a_k^j) \Rightarrow l_k \right]. \tag{3}$$

**Learning:** The TM adapts online using the training pair $(x, y)$. Tsetlin Automata (TAs) states are either incremented or decremented based on feedback, categorized as Type I (triggered when $y = 1$) and Type II (triggered when $y = 0$), shown in Table 1 and Table 2 respectively. For Type

| Input | Clause | 1 | | 0 | |
|---|---|---|---|---|---|
| | Literal | 1 | 0 | 1 | 0 |
| Include Literal | P(Reward) | $\frac{s-1}{s}$ | NA | 0 | 0 |
| | P(Inaction) | $\frac{1}{s}$ | NA | $\frac{s-1}{s}$ | $\frac{s-1}{s}$ |
| | P(Penalty) | 0 | NA | $\frac{1}{s}$ | $\frac{1}{s}$ |
| Exclude Literal | P(Reward) | 0 | $\frac{1}{s}$ | $\frac{1}{s}$ | $\frac{1}{s}$ |
| | P(Inaction) | $\frac{1}{s}$ | $\frac{s-1}{s}$ | $\frac{s-1}{s}$ | $\frac{s-1}{s}$ |
| | P(Penalty) | $\frac{s-1}{s}$ | 0 | 0 | 0 |

Table 1: Type I Feedback.

| Input | Clause | 1 | | 0 | |
|---|---|---|---|---|---|
| | Literal | 1 | 0 | 1 | 0 |
| Include Literal | P(Reward) | 0 | NA | 0 | 0 |
| | P(Inaction) | 1.0 | NA | 1.0 | 1.0 |
| | P(Penalty) | 0 | NA | 0 | 0 |
| Exclude Literal | P(Reward) | 0 | 0 | 0 | 0 |
| | P(Inaction) | 1.0 | 0 | 1.0 | 1.0 |
| | P(Penalty) | 0 | 1.0 | 0 | 0 |

Table 2: Type II Feedback.

I, when both clause and literal are 1-valued, it adjusts TA states upwards to capture patterns in $x$. When it is triggered for 0-valued clauses or literals, it adjusts states downwards to mitigate overfitting. For Type II, it targets the *Exclude* action to refine clauses, focusing on instances where the literal is 0-valued but its clause is 1-valued. The likelihood, governed by a user parameter $s > 1$, is mainly $\frac{s-1}{s}$ or $\frac{1}{s}$, but could be 1. The details of the updating rules can be found in Granmo (2018).

## 3 PCL: PROBABILISTIC INCLUSION OF LITERALS FOR CONCEPT LEARNING

In the PCL model, following the TM approach, a Tsetlin automata, denoted as $\text{TA}_j^i$, is associated with each literal $l_j$ and clause $C_i$ to decide the inclusion or exclusion of the literal $l_j$ in $C_i$. The target conjunction concept is represented by $C_T$ (e.g., $C_T = x_1 \wedge \neg x_2$). The number of literals in $C_T$ is given by $m = |C_T|$, with $m = 2$ for the given example. We say that the literal $l_j$ satisfies a sample $e$ (also denoted by $l_j \in e$) if it equals 1, and we say that $l_j$ violates a sample $e$ (also denoted by $l_j \notin e$) otherwise.

Given positive ($e^+$, i.e., $y = 1$) and negative ($e^-$, i.e., $y = 0$) samples, PCL learns a *disjunctive normal form* (DNF) formula: $C_1 \vee \cdots \vee C_k$ with $C_j = \bigwedge_{i=1}^{2n} \left[ g(a_i^j) \Rightarrow l_i \right]$. In PCL, this DNF formula classifies unseen samples, i.e., $\hat{y} = C_1 \vee \cdots \vee C_k$.

While both TM and PCL update the states of every TA using feedback from positive and negative samples, there are notable differences in the PCL approach. In PCL, the inaction transition is disabled, feedback is independent of the values of clauses during training, and instead of TM's uniform transition probabilities for each clause, PCL assigns a unique inclusion probability to each clause to diversify learned patterns.

Figure 2 provides an example of the PCL architecture with two clauses, each associated with a $p_i$ value. Black arrows represent *reward* transitions (enforce the action), whiles red arrows represent *penalty* transitions (penalize the action). The current TA states (represented by the black dots) translate to the clauses $C_1 = \neg x_1$ and $C_2 = \neg x_1 \wedge \neg x_2$. The DNF $C_1 \vee C_2$ can be then used to classify unseen samples. TA states are initialized randomly and updated based on sample feedback. Subsequent sections will delve into the feedback provided for each sample (on positive and negative samples respectively).

### 3.1 FEEDBACK ON POSITIVE SAMPLES

Table 3 details the feedback associated with a positive sample $e^+$. This feedback relies on the literal value and the current action of its corresponding TA. As an illustration, if a literal is 1 in a positive sample and the current action is "Include", then the reward probability is denoted by $p_i$, as shown in Table 3.

A notable distinction from TM is that PCL's feedback is independent of the clause's value. Instead, distinct probabilities are associated with each clause. As a result, Table 3 provides two columns: one for satisfied literals and another for violated literals. For instance, the positive sample $e^+(1, 0, 1, 0)$ satisfies literals $x_1$, $\neg x_2$, $x_3$, and $\neg x_4$ (having value 1) and violates literals $\neg x_1$, $x_2$, $\neg x_3$, and $x_4$ (having value 0).

| Action | Transitions | Feedback on $l_j \in e^+$ | Feedback on $\neg l_j \mid l_j \in e^+$ |
|--------|-------------|---------------------------|------------------------------------------|
| Include | $P(Reward)$ | $p_i$ | 0 |
|         | $P(Penalty)$ | $1 - p_i$ | 1 |
| Exclude | $P(Reward)$ | $1 - p_i$ | 1 |
|         | $P(Penalty)$ | $p_i$ | 0 |

Table 3: Feedback on a positive sample $e^+$ on clause $C_i$ with $p_i$.

| Action | Transitions | Feedback on $l_j \in e^-$ | Feedback on $\neg l_j \mid l_j \in e^-$ |
|--------|-------------|---------------------------|------------------------------------------|
| Include | $P(Reward)$ | $1 - p_i$ | $p_i$ |
|         | $P(Penalty)$ | $p_i$ | $1 - p_i$ |
| Exclude | $P(Reward)$ | $p_i$ | $1 - p_i$ |
|         | $P(Penalty)$ | $1 - p_i$ | $p_i$ |

Table 4: Feedback on a negative samples $e^-$ on clause $C_i$ with $p_i$.

When literals oppose the positive sample $e^+$ (refer to the last column of Table 3), it's evident that these literals are not constituents of the target conjunction. Therefore, they are excluded with a 1.0 probability. Conversely, literals aligning with $e^+$ (as shown in the third column of Table 3) don't have assured inclusion. For conjunctions with many literals, including the majority is favored. However, for those with a singular literal, the positive sample arises from one correct literal, rendering the others superfluous. In such scenarios, the inclusion of most literals is discouraged, by having a smaller $p_i$ value. Therefore, inclusion is activated based on a user-defined probability $p_i$, while exclusion relies on a probability of $1 - p_i$. This is further illustrated in Example 1.

### 3.2 FEEDBACK ON NEGATIVE SAMPLES

Table 4 delineates the feedback related to a negative sample $e^-$. Literals that violate the negative sample $e^-$ are potential candidates for inclusion. This is based on the rationale that negating certain literals might rectify the sample. However, the probability of their inclusion in clause $C_i$ is dictated by $p_i$, and their probable exclusion is initiated with a likelihood of $1 - p_i$ (refer to the last column of Table 4).

Conversely, literals that align with the negative sample $e^-$ are considered for exclusion, postulating that the literal's presence might be causing its negative label. Yet, this potential exclusion happens with a probability of $p_i$. Their probable inclusion is initiated with a likelihood of $1 - p_i$ (see the third column of Table 4).

**Example 1.** *Given $n = 4$, we plan to learn a target conjunction $C_T$. The set of possible literals are $x_1, x_2, x_3, x_4, \neg x_1, \neg x_2, \neg x_3$, and $\neg x_4$. If we have a positive sample, $e^+$, and we know that $C_T$ includes exactly 4 literals, then all literals satisfying $e^+$ should be included. However, if $C_T$ only includes one literal, we should include only one literal satisfying the sample $e^+$ (1 amongst 4 i.e., 25% of literals). Hence, we can control, to some extent, the size of the learned clause by setting the inclusion probability $p_i$ of the clause $C_i$ (large if we want more literals, small otherwise).*

### 3.3 PCL VS TM

When contrasting PCL with the standard TM, a primary distinction arises: In PCL, TAs for all literals within a clause are updated autonomously. This individualized treatment of literals simplifies the complexity of the theoretical convergence analysis compared to the traditional TM. Even with a nuanced learning approach, PCL's convergence proof offers profound insights into the learning concept, thereby enhancing the theoretical grasp of the TM family.

## 4 PCL CONVERGENCE PROOF

In this section, we prove that PCL almost surely converges to any conjunction of literals in infinite time horizon. Before that, we introduce some notations and definitions. There are three types of literals. We denote by $L_1$ literals $l_j$ such that $l_j \in C_T$ (correct literals), by $L_2$ literals $\neg l_j$ such that

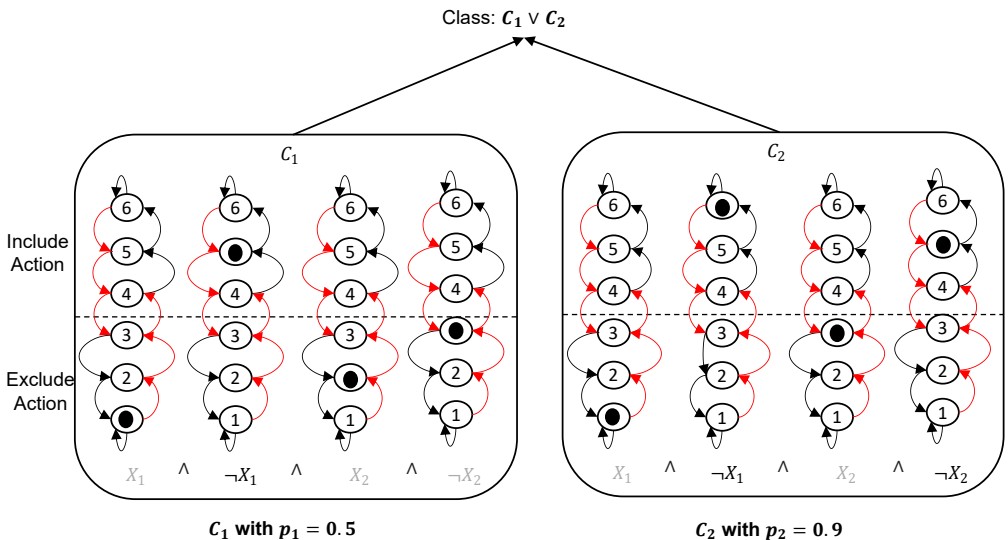

Figure 2: PCL example.

$l_j \in C_T$ (negative literals) and by $L_3$ literals $l_j$ such that $l_j \notin C_T \land \neg l_j \notin C_T$ (irrelevant literals). We now define the convergence to conjunction of literals.

**Definition 1** (Convergence to a conjunction of literals). *An algorithm almost surely convergences to a conjunction of literals $C_T$, if it includes every literal in $L_1$ and excludes every literal in $L_2$ and every literal in $L_3$ in infinite time horizon.*

Given all the possible $2^n$ samples, and a literal $l_j$, we distinguish four sample classes:

$A_1$: Positive samples, $e^+$, such that $l_j$ is satisfied ($l_j \in e^+$).

$A_2$: Positive samples, $e^+$, such that $l_j$ is violated ($l_j \notin e^+$).

$A_3$: Negative samples, $e^-$, such that $l_j$ is satisfied ($l_j \in e^-$).

$A_4$: Negative samples, $e^-$, such that $l_j$ is violated ($l_j \notin e^-$).

In Table 5, we report the number of samples for each class/literal type. We denote by $freq(i,j)$ the number of samples for a literal of type $L_i$ in class $A_j$ and by $\alpha_{i,j}$ the relative frequency i.e., $\alpha_{i,j} = \frac{freq(i,j)}{2^n}$. For example, $freq(1,4) = 2^{n-1}$ and $\alpha_{1,4} = \frac{2^{n-1}}{2^n} = 0.5$.

**Example 2.** *Suppose that $n = 3$ and $C_T = x_1 \land \neg x_2$. We have $2^3$ possible samples, i.e., $(0,0,0)$, $(0,0,1)$, $(0,1,0)$, $(1,0,0)$, $(0,1,1)$, $(1,0,1)$, $(1,1,0)$, $(1,1,1)$. Following $C_T$, there are $2^{3-2} = 2$ positive samples and $2^3 - 2^{3-2} = 6$ negative samples. Literals in $L_1$ are $x_1$ and $\neg x_2$, literals in $L_2$ are $\neg x_1$ and $x_2$ and literals in $L_3$ are $x_3$ and $\neg x_3$. For example, given the literal $x_1$ (in $L_1$): samples in class $A_1$ (positive samples where $x_1$ equals 1) are $(1,0,0)$ and $(1,0,1)$; it is impossible to have a sample in class $A_2$, i.e, $freq(1,2) = 0$ (we cannot have a positive sample that violates a literal in $C_T$); samples in class $A_3$ (negative samples where $x_1$ equals 1) are $(1,1,0)$ and $(1,1,1)$; samples in class $A_4$ (negative samples where $x_1$ equals 0) are $(0,0,0)$, $(0,0,1)$, $(0,1,0)$ and $(0,1,1)$. Note that in the special case, where $m = n$, there are no literals in $L_3$.*

Following feedback tables, Table 6 presents possible actions and probability associated to each action for each class of samples for a clause $C_k$ (with inclusion probability $p_k$).

**Lemma 1.** *For $p$ and $\alpha$ between 0 and 1, $\alpha \times p + (1 - \alpha) \times (1 - p) > 0.5$ if and only if $(p > 0.5 \land \alpha > 0.5)$ or $(p < 0.5 \land \alpha < 0.5)$.*

**Theorem 1.** *Given a PCL with one clause, $C_k$, PCL will almost surely converge to the target conjunction $C_T$ in infinite time horizon if $0.5 < p_k < 1$.*

| Literal | $A_1$ | $A_2$ | $A_3$ | $A_4$ |
|---|---|---|---|---|
| $L_1$ | $2^{n-m}$ | 0 | $2^n - 2^{n-m} - 2^{n-1}$ | $2^{n-1}$ |
| $L_2$ | 0 | $2^{n-m}$ | $2^{n-1}$ | $2^n - 2^{n-m} - 2^{n-1}$ |
| $L_3$ | $2^{n-m-1}$ | $2^{n-m-1}$ | $2^{n-1} - 2^{n-m-1}$ | $2^{n-1} - 2^{n-m-1}$ |

Table 5: Frequency of each class w.r.t each literal type.

| Sample Class | Actions | Probability |
|---|---|---|
| $A_1$ | Include | $p_k$ |
|  | Exclude | $1 - p_k$ |
| $A_2$ | Include | 0.0 |
|  | Exclude | 1.0 |
| $A_3$ | Include | $1 - p_k$ |
|  | Exclude | $p_k$ |
| $A_4$ | Include | $p_k$ |
|  | Exclude | $1 - p_k$ |

Table 6: Summary of feedback w.r.t each sample class.

*Proof.* Given a clause $C_k$ with inclusion probability $p_k$, we prove that $C_k$ converges to the target $C_T$. In other words, we prove that every literal in $L_1$ is included and every literal in $L_2$ or $L_3$ is excluded in infinite time horizon.

Given a clause $C_k$, we suppose that every $\text{TA}_j^k$ has a memory size of $2N$. Let $I_k(l_j)$ denote the action Include (when $g(a_j^k)$ is 1) and $E_k(l_j)$ the action Exclude (when $g(a_j^k)$ is 0) and let $a_j^k(t)$ denote the state of $\text{TA}_j^k$ at time $t$. Because we have a single clause, $C_k$, the index $k$ can be omitted.

Let $E(l_j)^+$ (resp. $I(l_j)^+$) denote the event where $E(l_j)$ is reinforced (resp. $I(l_j)$ is reinforced), meaning a transition towards the most internal state of the arm, namely, state 1 for action $E(l_j)$ (resp. state $2N$ for action $I(l_j)$).

If $a_j \leq N$, the action is $E(l_j)$. $E(l_j)^+$ corresponds to a reward transition, meaning, the $\text{TA}_j$ translates its state to $a_j - 1$ unless $a_j = 1$, then it will maintain the current state. In other terms, at time $t$, $P(E(l_j)^+)$ is $P(a_j(t+1) = f - 1 \mid a_j(t) = f)$ if $1 < f \leq N$. In case, $f = 1$, $P(E(l_j)^+)$ is $P(a_j(t+1) = 1 \mid a_j(t) = 1)$.

If $a_j > N$, the action is $I(l_j)$. $I(l_j)^+$ corresponds to a reward transition, meaning, $\text{TA}_j$ translates its state to $a_j + 1$ unless $a_j = 2N$, then it will maintain the current state. In other terms, at time $t$, $P(I(l_j)^+)$ is $P(a_j(t+1) = f + 1 \mid a_j(t) = f)$ if $N \leq f < 2N$. In case, $a_j = 2N$, $P(I(l_j)^+)$ is $P(a_j(t+1) = 2N \mid a_j(t) = 2N)$.

We distinguish three cases: $l_j \in L_1$, $l_j \in L_2$ and $l_j \in L_3$.

**Case 1:** $l_j \in L_1$   We will prove $P(I(l_j)^+) > 0.5 > P(E(l_j)^+)$.

Following Tables 5 and 6, we have:

$$P(I(l_j)^+) = \alpha \times p_k + (1 - \alpha) \times (1 - p_k) \ s.t. \ \alpha = \alpha_{1,1} + \alpha_{1,4}.$$

This is, samples in classes $A_1$ and $A_4$ ($\alpha$ of the samples) suggest to include with probability $p_k$ and samples in $A_3$ ($(1 - \alpha)$ of the samples) suggest to include with probability $(1 - p_k)$. We know that $\alpha_{1,4} = \frac{2^{n-1}}{2^n} = 0.5$, hence:

$$\alpha_{1,1} + \alpha_{1,4} > 0.5, \ i.e., \ \alpha > 0.5.$$

Supposing that $p_k > 0.5$, then we have $P(I(l_j)^+) > 0.5$ (following Lemma 1).

We now compute $P(E(l_j)^+)$:

$$P(E(l_j)^+) = \alpha \times (1 - p_k) + (1 - \alpha) \times p_k \ s.t. \ \alpha = \alpha_{1,1} + \alpha_{1,4}.$$

This is, samples in classes $A_1$ and $A_4$ ($\alpha$ of the samples) suggest to exclude with probability $(1 - p_k)$ and samples in class $A_3$ ($(1 - \alpha)$ of the samples) suggest to include with probability $p_k$.

From before, we know that $\alpha > 0.5$ and $(1-p_k) < 0.5$, then $P(E(l_j)^+) < 0.5$ (following Lemma 1). Then:

$$P(I(l_j)^+) > 0.5 > P(E(l_j)^+).$$

Thus, **literals in $L_1$ will almost surely be included in infinite time horizon if $p_k > 0.5$.**

**Case 2:** $l_j \in L_2$    We will prove $P(E(l_j)^+) > 0.5 > P(I(l_j)^+)$.

Following Tables 5 and 6, we have:

$$P(E(l_j)^+) = \alpha_{2,2} \times 1.0 + \alpha_{2,3} \times p_k + \alpha_{2,4} \times (1 - p_k).$$

This is, samples in class $A_2$ ($\alpha_{2,2}$ of the samples) suggest to exclude with probability 1.0, samples in $A_3$ ($\alpha_{2,3}$ of the samples) suggest to exclude with probability $p_k$ and samples in $A_4$ ($\alpha_{2,4}$ of the samples) suggest to exclude with probability $(1 - p_k)$. We know that:

$$\alpha_{2,3} = 0.5 \implies \alpha_{2,2} + \alpha_{2,3} > 0.5,$$

$$\alpha_{2,2} + \alpha_{2,3} + \alpha_{2,4} = 1 \implies \alpha_{2,4} = 1 - \alpha_{2,2} - \alpha_{2,3}.$$

Because $p_k$ is a probability, we know that $\alpha_{2,2} \times 1.0 \geq \alpha_{2,2} \times p_k$, hence:

$$\alpha_{2,2} \times 1.0 + \alpha_{2,3} \times p_k \geq \alpha_{2,2} \times p_k + \alpha_{2,3} \times p_k$$
$$\implies \quad \alpha_{2,2} \times 1.0 + \alpha_{2,3} \times p_k \geq (\alpha_{2,2} + \alpha_{2,3}) \times p_k.$$
$$\implies \quad \alpha_{2,2} \times 1.0 + \alpha_{2,3} \times p_k + \alpha_{2,4} \times (1 - p_k) \geq (\alpha_{2,2} + \alpha_{2,3}) \times p_k + \alpha_{2,4} \times (1 - p_k)$$
$$\implies \quad P(E(l_j)^+) \geq (\alpha_{2,2} + \alpha_{2,3}) \times p_k + (1 - \alpha_{2,2} - \alpha_{2,3}) \times (1 - p_k).$$

Knowing that $\alpha_{2,2} + \alpha_{2,3} > 0.5$, supposing that $p_k > 0.5$, and following Lemma 1, we have:

$$(\alpha_{2,2} + \alpha_{2,3}) \times p_k + (1 - \alpha_{2,2} - \alpha_{2,3}) \times (1 - p_k) > 0.5 \implies P(E(l_j)^+) > 0.5.$$

We now compute $P(I(l_j)^+)$:

$$P(I(l_j)^+) = \alpha_{2,3} \times (1 - p_k) + \alpha_{2,4} \times p_k.$$

This is, samples in class $A_3$ ($\alpha_{2,3}$ of the samples) suggest to include with probability $(1 - p_k)$ and samples in $A_4$ ($\alpha_{2,4}$ of the samples) suggest to include with probability $p_k$.

We know that $\alpha_{2,3} = \frac{2^{n-1}}{2^n} = 0.5$ and $\alpha_{2,4} < 0.5$, then:

$$\alpha_{2,4} \times p_k < 0.5 \times p_k$$
$$0.5 \times (1 - p_k) + \alpha_{2,4} \times p_k < 0.5 \times (1 - p_k) + 0.5 \times p_k,$$
$$0.5 \times (1 - p_k) + \alpha_{2,4} \times p_k < 0.5 \implies P(I(l_j)^+) < 0.5.$$

Then:

$$P(E(l_j)^+) > 0.5 > P(I(l_j)^+).$$

Thus, **literals in $L_2$ will almost surely be excluded in infinite time horizon if $p_k > 0.5$.**

**Case 3:** $l_j \in L_3$    We will prove $P(E(l_j)^+) > 0.5 > P(I(l_j)^+)$.

Following Tables 5 and 6, we have:

$$P(E(l_j)^+) = (\alpha_{3,1} + \alpha_{3,4}) \times (1 - p_k) + \alpha_{3,2} \times 1.0 + \alpha_{3,3} \times p_k.$$

This is, samples in classes $A_1$ and $A_4$ ($\alpha_{3,1} + \alpha_{3,4}$ of the samples) suggest to exclude with probability $(1 - p_k)$, samples in $A_2$ ($\alpha_{3,2}$ of the samples) suggest to exclude with probability 1.0 and samples in $A_3$ ($\alpha_{3,3}$ of the samples) suggest to exclude with probability $p_k$.

From Table 5, we know that $\alpha_{3,1} = \alpha_{3,2}$ and $\alpha_{3,3} = \alpha_{3,4}$. Hence, by substitution:

$$P(E(l_j)^+) = (\alpha_{3,1} + \alpha_{3,3}) \times (1 - p_k) + \alpha_{3,1} + \alpha_{3,3} \times p_k,$$

$$P(E(l_j)^+) = \alpha_{3,1} \times (1 - p_k) + \alpha_{3,3} \times (1 - p_k) + \alpha_{3,1} + \alpha_{3,3} \times p_k,$$

$$P(E(l_j)^+) = \alpha_{3,1} \times (2 - p_k) + \alpha_{3,3}.$$

We add and remove $\alpha_{3,1}$:

$$P(E(l_j)^+) = \alpha_{3,1} \times (2 - p_k) + \alpha_{3,3} + \alpha_{3,1} - \alpha_{3,1}.$$

From Table 5, we know that:

$$\alpha_{3,3} + \alpha_{3,1} = 0.5.$$

Then:

$$P(E(l_j)^+) = \alpha_{3,1} \times (2 - p_k) + 0.5 - \alpha_{3,1} \implies P(E(l_j)^+) = \alpha_{3,1} \times (1 - p_k) + 0.5.$$

We want to prove that:

$$\alpha_{3,1} \times (1 - p_k) + 0.5 > 0.5.$$

That is true only if:

$$\alpha_{3,1} \times (1 - p_k) > 0.$$

Given that $L_3 \neq \emptyset$, we know that $\alpha_{3,1} > 0$, then we need:

$$1 - p_k > 0 \implies p_k < 1.$$

Hence, $P(E(l_j)^+) > 0.5$, but only if $p_k < 1$.

We now compute $P(I(l_j)^+)$:

$$P(I(l_j)^+) = (\alpha_{3,1} + \alpha_{3,4}) \times p_k + \alpha_{3,2} \times 0.0 + \alpha_{3,3} \times (1 - p_k).$$

$$P(I(l_j)^+) = (\alpha_{3,1} + \alpha_{3,4}) \times p_k + \alpha_{3,3} \times (1 - p_k).$$

This is, samples in classes $A_1$ and $A_4$ ($\alpha_{3,1} + \alpha_{3,4}$ of the samples) suggest to include with probability $p_k$ and samples in $A_3$ ($\alpha_{3,3}$ of the samples) suggest to include with probability $(1 - p_k)$. From Table 5, we know that $\alpha_{3,1} + \alpha_{3,4} = 0.5$ and $\alpha_{3,3} < 0.5$, then:

$$0.5 \times p_k + \alpha_{3,3} \times (1 - p_k) < 0.5 \times p_k + 0.5 \times (1 - p_k)$$

$$0.5 \times p_k + \alpha_{3,3} \times (1 - p_k) < 0.5 \implies P(I(l_j)^+) < 0.5$$

$$P(E(l_j)^+) > 0.5 > P(I(l_j)^+).$$

Thus, **literals in $L_3$ will almost surely be excluded in infinite time horizon if** $p_k < 1$.

Hence, following Definition 4, PCL will almost surely converge to $C_T$ in infinite time horizon if $0.5 < p_k < 1$.

$\square$

## 5   EXPERIMENTAL EVALUATION

Having established the theoretical convergence of PCL in Theorem 1, this section aims to substantiate these findings through empirical tests.[1] Our experimental goal can be succinctly framed as:

> *"Given all possible combinations as training examples (i.e., $2^n$ samples), if we run PCL (with a single clause $C_k$ and inclusion probability $p_k$) 100 times — each time with a distinct, randomly generated target conjunction $C_T$ and varying number of epochs — how often does PCL successfully learn the target conjunction $C_T$?"*

---

[1]Code is available at https://anonymous.4open.science/r/dpcl-classifier-6D92/README.md

### 5.1 EXPERIMENT 1: FIXED INCLUSION PROBABILITY (FIGURE 3(A))

In this experiment, the inclusion probability $p_k$ is set to a fixed value of 0.75, which falls between 0.5 and 1.0. We then observe the number of successful learnings in relation to the number of epochs for different feature counts, denoted as $n$. We observe the following:

- As the epoch count increases, PCL consistently converges to the target, achieving this 100% of the time for larger epoch numbers.
- For $n = 8$, PCL identifies all 100 targets by the 1,000th epoch. Furthermore, even at 600 epochs, the success rate is over 90%.

This performance corroborates the assertions made in Theorem 1, suggesting PCL's capability to converge over an infinite time horizon.

### 5.2 EXPERIMENT 2: VARYING INCLUSION PROBABILITIES (FIGURE 3(B))

Here, we keep the feature count fixed at $n = 4$ and vary the inclusion probability $p_k$. The number of successful learnings is plotted against the number of epochs. We observe the following:

- For $p_k < 0.5$, PCL struggles to learn the targets, with success rates close to zero across epochs.
- On the contrary, for $0.5 < p_k < 1$, PCL shows remarkable improvement, especially as the number of epochs increases.
- Interestingly, at $p_k = 1$, the success rate diminishes, indicating non-convergence.

These outcomes validate our precondition for PCL's convergence: $0.5 < p_k < 1$. Overall, the findings from our experiments robustly support Theorem 1, emphasizing that PCL exhibits convergence within the range $0.5 < p_k < 1$.

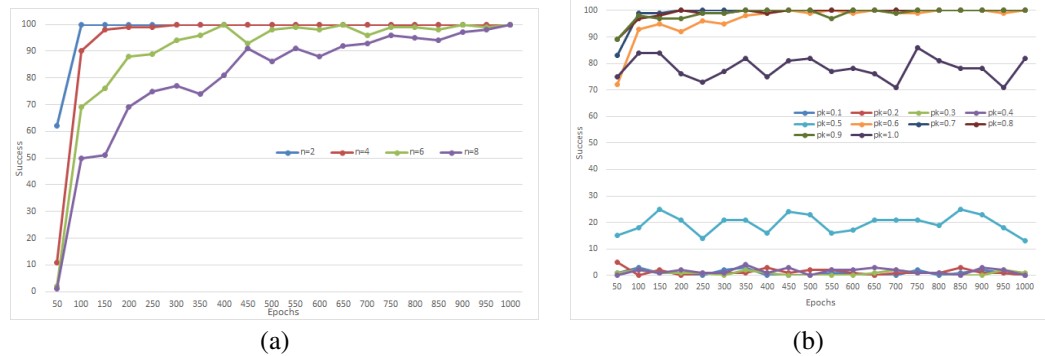

(a)                   (b)

Figure 3: (a) The number of successes for each number of features $n$ w.r.t the number of epochs with $p_k = 0.75$. (b) The number of successes w.r.t the number of epochs for $n = 4$ and different $p_k$ values.

## 6 CONCLUSION

In this study, we introduced PCL, an innovative Tsetlin Agent-based method for deriving Boolean expressions. Distinctively, PCL streamlines the TM training process by integrating dedicate inclusion probability to each clause, enriching the diversity of patterns discerned. A salient feature of PCL is its proven ability to converge to any conjunction of literals over an infinite time horizon, given certain conditions—a claim corroborated by our empirical findings. This pivotal proof lays a solid foundation for the convergence analysis of the broader TM family. The implications of our theoretical insights herald potential for PCL's adaptability to real-world challenges. Looking ahead, our ambition is to enhance PCL's capabilities to cater to multi-class classification and to rigorously test its efficacy on practical applications.

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

| Methods | Binary Iris | Breast Cancer |
|---|---|---|
| Naive Bayes | 91.6 | 64.2 |
| Logistic Regression | 92.6 | 65.5 |
| 1-layer NN | 93.8 | 71.9 |
| SVM | 93.6 | 67.8 |
| DT | 94.7 | 70.6 |
| RF | 95.5 | 74.7 |
| KNN | 91.1 | 75.5 |
| TM (300 clauses) | 95.0 | 70.6 |
| PCL (10 clauses) | 94.2 | 69.5 |

Table 7: Empirical performance comparison.

| Methods | n=6 | n=10 |
|---|---|---|
| PCL (2 clauses) | 100 | 100 |
| TM (2 clauses) | 96.87 | 93.50 |

Table 8: Noise-free example.

---

**Algorithm 1** PCL training

---

1: **Input:** $b$ training examples $(e_j, y_j)$
2: **Initialize:** Random initialization of TAs
3: **Begin:** $n^{th}$ training round
4: **for** $e_j \in \{e_1, ..., e_b\}$ **do**
5:     **for** $C_i \in \{C_1, ..., C_m\}$ **do**
6:         **if** $(y_j = 1)$ **then**
7:             **for** $(l_k \in e_j$ and $l_k \notin e_j)$ **do**
8:                 Feedback on positive samples w.r.t $p_i$
9:             **end for**
10:         **else**: $(y_j = 0)$
11:             **for** $(l_k \in e_j$ and $l_k \notin e_j)$ **do**
12:                 Feedback on negative samples w.r.t $p_i$
13:             **end for**
14:         **end if**
15:     **end for**
16: **end for**

---

## A    PCL AS CLASSIFIER

In our study, we employed PCL as a classification tool, using its Disjunctive Normal Form (DNF) output, and tested it on binary iris and breast cancer datasets. We compared PCL's performance with that of the vanilla Tsetlin Machine (TM) and various established machine learning algorithms, with results detailed in Table 7. Notably, PCL achieved competitive results with just 10 clauses over 50 training epochs, setting clause probabilities uniformly in the range [0.6, 0.8]. For comparison, other methods used default settings from their implementations, while TM used 300 clauses with specific settings ($s = 2$, $T = 10$) over 100 epochs. PCL, with the right $p_i$ settings, also attained a 96% accuracy rate on a binary MNIST dataset, distinguishing between 0s and 1s.

Additionally, we analyzed the convergence of PCL and TM using deterministic sample data, consisting of $n$ literals and a fixed target expression. We used this data to train and test the models, selecting samples from all possible combinations. Both PCL and TM were limited to two clauses. As shown in Table 8, PCL achieved 100% accuracy aligning with our convergence proof. In contrast, TM reached 96.87% and 93.50% accuracy for $n = 6$ and $n = 10$, respectively. It is important to note that these accuracy figures, presented in Tables 7 and 8, are averages from 10 independent runs.

## B    PCL ALGORITHM

A formal algorithmic pseudo-code for PCL is given in Algorithm 1

