# OpenReview forum: "Generalized Convergence Analysis of Tsetlin Machines: A Probabilistic Approach to Concept Learning"
_ICLR.cc/2024/Conference — Submitted to ICLR 2024_

### Official Review · Reviewer_Q3AU · 2023-11-01

**Soundness:** 2 fair
**Presentation:** 2 fair
**Contribution:** 3 good
**Rating:** 5
**Confidence:** 3

**Summary:**

Tsetlin Machines are an automata theoretic formalism for the generation of Boolean expressions as conjunctive clauses. Proofs of convergence for Tsetlin machines have been provided in the cases of 1 and 2 input bits. Such a proof of convergence remains elusive in the general case. In particular, for any number of bits greater than 2.

In this work the authors provide a probabilistic variant of Tsetlin machines. They call their novel framework PCL (Probabilistic Concept Learning). The main result of the paper is a proof that for any clause C_k, PCL converges almost surely to a conjunction of minerals whenever the inclusion probability p_k lies strictly between 0.5 and 1.

**Strengths:**

The main strength of the paper is the fact that in the new introduced model, the authors can provide convergence guarantees in the general case, provided the inclusion probability of the clause in question lies between 0.5 and 1.0.

**Weaknesses:**

The main weakness of the paper is in the wording of some claims. For example already in the abstract, the authors write:

"Despite this, the convergence proof for the TMs, ... remains an open problem. This paper aims to fill this gap by presenting a comprehensive convergence analysis of Tseitin automaton-based Machine learning algorithms".

This is an extremely unclear way of formulating your contributions. First, it gives the impression that you are solving the open problem that remained open about the convergence of Tsetlin machines in the general case. On the other hand, the convergence claims that you explicit mention are with respect to another model (PCL).

**Questions:**

Does the proof of convergence for your model PCL imply a proof of convergence for the original TA model in the general case?

1) If yes, please state this very explicitly in the abstract, and introduction. Something like: "We note that our proof of convergence implies a proof of convergence for TA's". Therefore, our proof of convergence settles the long-standing open problem about convergence of TAs for number of bits greater thant 2.
2) If no, please state this very explicitly in the abstract and introduction. Something like: "We note that the original problem about the convergence of TA's remains widely open for the case of number of bits greater than 2."

---

> ### Author Response · Authors · 2023-11-14
> **Response to Reviewer Q3AU**
>
> Thank you for your constructive feedback. As mentioned in our general response, Tsetlin Automatons serve as foundational components for both PCL and the original Tsetlin Machine (TM), but they should not be mistaken for the Tsetlin Machine approach detailed in Section 2. It is important to note that our proof is the first of its kind for approaches utilizing Tsetlin Automatons, such as TM and PCL, but it does not apply to the original Tsetlin Machine model itself.
> We have updated the title, the abstract and added a sentence at the end of the introduction to remove any confusion.
> We thank the reviewer again for pointing this our.

---

### Official Review · Reviewer_aoEu · 2023-11-01

**Soundness:** 2 fair
**Presentation:** 1 poor
**Contribution:** 2 fair
**Rating:** 3
**Confidence:** 3

**Summary:**

This paper proposes a novel ML model, dubbed Probabilistic inclusion of literals for Concept Learning (PCL).
PCL modifies Tsetlin Machines (TMs) in order to provide theoretical convergence guarantees in ideal settings.
This property is further verified experimentally on synthetic data.

**Strengths:**

- In contrast with the original TMs, PCL comes with theoretical guarantees of convergence

**Weaknesses:**

- For non-experts in TMs, the paper is hardly accessible. I am having an hard time understanding the background (see detailed feedback)
- The theoretical guarantee hold in idealized cases and relies on the novel PCL formulation, it is not clear whether it extends to TMs in general as the title would suggest.
- Very weak evaluation: experiments only corroborate the theoretical proofs. PCL is not compared with any existing method for learning a logical formulas, not even standard TMs. The significance of this theoretical results depends on whether PCL is a practical ML algorithm or not.
- The authors mention practical implications of this work in the abstract. I couldn't find these implications in the text.


Detailed feedback:
----

The background assumes that the reader knows what a Tsetlin Automata is.


  "Clause updating probability depends on vote sum, v, and the voting error, ε, uses a margin T to produce an ensemble effect"

This is the first time a sentence implies that feedback might be disregarded. Vote sum, vote error and the margin are mentioned for the first and only time.

  "Upon sampling from P(Feedback), TA state adjustments are computed "

How does the sampling work? Input pairs (x,y) are sampled according to P(Feedback), which in turn depends on the prediction of the current TM state? This should be clarified.

I didn't get the need for differentiating Type Ia and Ib feedback. This is never used later in the text.

**Questions:**

- Is PCL a strict generalization of TM? Do the theoretical guarantees hold for TMs in general?
- Is PCL accurate in practical settings? How do they compare with standard TMs and other concept learning algorithms?
- What are the practical implications of this work?

---

> ### Author Response · Authors · 2023-11-14
> **Response to Reviewer aoEu**
>
> Thank you for your constructive feedback. Section 2 of our paper provides a description of the original Tsetlin Machine, which should not be confused with the Tsetlin Automaton. It is true that some concepts introduced there are exclusive to the original Tsetlin Machine and do not apply to the basic framework of PCL. The reviewer's concern about the unclear description of sampling has been resolved by removing the relevant content from the paper. The parameters involved in sampling, including the vote sum ($v$), the voting error ($\epsilon$), and the margin ($T$), are not employed in either the PCL method or its proof. These parameters are specifically used in vanilla TM for differentiating between various sub-patterns. To prevent confusion, we have decided to omit certain notations that are irrelevant to PCL and will focus only on defining the key aspects of the original Tsetlin Machine (check our updated Section 2). We thank the reviewer for pointing this out.
>
> For information on the practical implications, results, and the generalization of the PCL proof, please refer to the answer provided in the general response section.

---

> > ### Comment · Reviewer_aoEu · 2023-11-23
> > **Response to authors**
> >
> > Thank you for the response. It is now clear that the theoretical results on the convergence of PCL do not apply to TMs.
> > This makes a throughout evaluation of the PCL model a necessity. The additional experiments is an initial step towards that goal but, in my opinion, they are not sufficient to convince the general audience that PCL is a promising learning framework.
> > I would expect more experiments, mean and standard deviation of the accuracy over repeated runs, training runtime.
> > This of course require some time and effort and in my opinion they should be the priority before deriving any theoretical convergence result. Ultimately, it is not clear to me how significant are the theoretical contributions and how they could benefit other models.

---

> > > ### Author Response · Authors · 2023-11-23
> > > **Importance of the convergence proof**
> > >
> > > We express our gratitude to the reviewer for the insightful comment.
> > >
> > > Emphasizing the significance of the convergence proof, we underscore its role in offering **valuable guidelines and essential elements** for proving convergence in approaches using Tsetlin Automatons as foundational components.
> > >
> > > Our contribution ensures that future analyses of convergence in Tsetlin Automaton-based machine learning approaches need not to start from scratch, instead, researchers can leverage the insights provided in our proof.
> > >
> > > Furthermore, our proof holds potential utility in the convergence analysis of the original Tsetlin Machine model.
> > > A best practice in tackling a complex problem (the convergence analysis of the Tsetlin Machine in this case) is to initiate the inquiry with a less complex, but similar, problem (the convergence proof of PCL).
> > > We are optimistic that **our proof will serve as a valuable resource for the Tsetlin machine community**, aiding in the examination of convergence across various variants.
> > >
> > > We extend our gratitude once again to the reviewer for his thoughtful consideration and time. We kindly request a reconsideration of the evaluation, confident that our work will contribute to the advancement of knowledge in this domain.

---

### Official Review · Reviewer_TRnW · 2023-11-02

**Soundness:** 3 good
**Presentation:** 3 good
**Contribution:** 2 fair
**Rating:** 6
**Confidence:** 3

**Summary:**

This is a theoretical study of a simple machine learning model. The setting is to learn a conjunction of boolean signals. The model is simple and clean enough for theoretical analysis and the result is non-trivial: In their model, any conjunction can be learned eventually, given minimal assumptions about the input distribution. The main result is not surprising to me, but it may help to further understand their learning approach.

**Strengths:**

Clean theory and presentation. A small experimental section emphasize the main result.

**Weaknesses:**

There is no clear impact on applications.

I am also missing a discussion how this model compares to other learning representations. The authors also missed to run some experiments to analyze whether this approach could work for more advanced cases (like learning two or more clauses, perhaps even real data).

**Questions:**

How does this model compare to other popular learning representations? To me, this model looks almost like learning a matrix of quantized values, but with some additional constrains.

What exactly is an epoch in your setting?

---

> ### Author Response · Authors · 2023-11-14
> **Response to Reviewer TRnW**
>
> Thank you for your constructive feedback. Our article primarily contributes to the theoretical realm, specifically impacting the family of approaches based on the Tsetlin automaton. Additionally, the convergence proof we present is of significant importance; it assures that, eventually, PCL will be able to learn an accurate hypothesis, meaning it will either exactly match or closely approximate the solution needed for practical problems. For more details on this, please refer to the experimental results in our updated paper and the general response section.
>
> Also, to clarify, an 'epoch' in our context refers to a single cycle where feedback is provided for all training examples.

---

### Official Review · Reviewer_qHKG · 2023-11-03

**Soundness:** 3 good
**Presentation:** 3 good
**Contribution:** 3 good
**Rating:** 5
**Confidence:** 2

**Summary:**

This paper describes an adaptation of the Tsetlin automata that simplify the structure  but allows different parameters for literals and ensures convergence.

**Strengths:**

Converges

**Weaknesses:**

but is the price ok?

**Questions:**

I found the paper to be mostly well written.

Introduction: a few details were discussed before they were introduced.

You mention that your results require a conjunction of literals. This seems quite restrictive to me?

Sec 2: this is a nice short intro to the Tsetln machines. I would have liked more state-of-the-art, and namely examples justifying the need for convergence (some algs such as loopy message passing often work well without providing guarantees they converge)

Sec 3: you introduce your alg piecemeal, as a  discussion, I would also like to see a more self contained even formal description.

Fig 2 might benefit from more description,

I could not verify sec 4

sec5: why not compare with std Tsetln?

---

> ### Author Response · Authors · 2023-11-14
> **Response to Reviewer qHKG**
>
> Thank you for your constructive feedback. We acknowledge that solely learning a single conjunction of literals might appear limited. However, in the context of PCL, it is possible to learn multiple clauses. These conjunctions act as fundamental elements that are then aggregated to develop a comprehensive Disjunctive Normal Form (DNF).
>
> We concur that proving the convergence of a machine learning model is essential, as it affirms the model's reliability and stability, ensuring consistent and dependable solutions. This proof not only aids in the development and evaluation of algorithms but also serves as a theoretical benchmark for assessing performance and comprehending the limitations of the model. It is true that some algorithms may perform effectively in practical applications even without a formal proof of convergence. Nonetheless, providing such proof undeniably enhances the robustness of the approach. To support this, we have included relevant references in the second paragraph of our introduction.
>
> For a more detailed and formal description of PCL, we provided an algorithm in the appendix of our paper. Additionally, we have improved the description of Figure 2 for better clarity.
> For additional experiments, please
> refer to the experimental results in our updated paper and the general response section.

---

### Author Response · Authors · 2023-11-14
**General Answer**

We thank the reviewers for their thorough and helpful feedback.

**Main Contribution of the paper:**
Our paper is fundamentally theoretical, focusing on providing the first ever general convergence proof for approaches based on the Tsetlin Automaton. This proof not only demonstrates convergence for automaton-based methods but also lays the groundwork for potentially establishing a general convergence proof for the original Tsetlin Machine.

It is important to clarify that our convergence proof specifically pertains to the Tsetlin Automaton family, not the original Tsetlin Machine model itself. Currently, a comprehensive proof for the original Tsetlin Machine remains elusive, primarily due to the complex interplay between literals. Reviewer Q3AU highlighted a possible point of confusion, which we address here: Tsetlin Automatons [Tsetlin (1961)], which are fundamental components of both PCL and the original Tsetlin Machine (as detailed in Section 2), should not be mistaken for the Tsetlin Machine approach itself [Granmo (2018)].

**Usefulness of the contribution and practical implications:**
Establishing convergence in a machine learning model is essential because it ensures the model's reliability and stability. Specifically, in our context, proving convergence means that PCL will eventually learn an accurate hypothesis, which translates to an exact or nearly exact solution for practical problems.

**Experimental evaluation:**
Many reviewers noted that our paper lacks comprehensive experimental evaluation, particularly comparisons with state-of-the-art methods. We initially omitted detailed experimental results to maintain focus on our main contribution, the convergence proof. Nevertheless, in response to these requests, we have included some preliminary findings below and in the updated PDF. Currently, PCL functions as a binary classifier; evolving it into a multiclass classifier will necessitate changes in our implementation, which we are actively working on as time permits.

**PCL as classifier:**

In our study, we employed PCL as a classification tool, using its Disjunctive Normal Form (DNF) output, and tested it on binary iris and breast cancer datasets. We compared PCL's performance with that of the vanilla Tsetlin Machine (TM) and various established machine learning algorithms, with results detailed in Table 1. Notably, PCL achieved competitive results with just 10 clauses over 50 training epochs, setting clause probabilities uniformly in the range $[0.6, 0.8]$. For comparison, other methods used default settings from their sklearn implementations, while TM used $300$ clauses with specific settings $(s=2, T=10)$ over $100$ epochs. PCL, with the right $p_i$ settings, also attained a 96%  accuracy rate on a binary MNIST dataset, distinguishing between 0s and 1s.

Additionally, we analyzed the convergence of PCL and TM using deterministic sample data, consisting of $n$ literals and a fixed target expression. We used this data to train and test the models, selecting samples from all possible combinations. Both PCL and TM were limited to two clauses. As shown in Table 2, PCL achieved 100\% accuracy aligning with our convergence proof. In contrast, TM reached 96.87\% and 93.50\% accuracy for $n=6$ and $n=10$, respectively. It is important to note that these accuracy figures, presented in Tables 1 and 2, are averages from $10$ independent runs.

| **Methods**            | **Binary Iris** | **Breast Cancer** |
|------------------------|------------------|-------------------|
| Naive Bayes            | 91.6             | 64.2              |
| Logistic Regression    | 92.6             | 65.5              |
| 1-layer NN             | 93.8             | 71.9              |
| SVM                    | 93.6             | 67.8              |
| DT                     | 94.7             | 70.6              |
| RF                     | 95.5             | 74.7              |
| KNN                    | 91.1             | 75.5              |
| TM (300 clauses)       | 95.0             | 70.6              |
| PCL (10 clauses)       | 94.2             | 69.5              |

*Table 1: Empirical performance comparison.*

|                       | **n=6** | **n=10** |
|-----------------------|-----------------|------------------|
| PCL (2 clauses)       | 100             | 100              |
| TM (2 clauses)        | 96.87           | 93.50            |

*Table 2: Noise-free example with deterministic target*


 **[Tsetlin (1961)]** Michael Lvovitch Tsetlin. On Behaviour of Finite Automata in Random Medium. Avtomat. i
Telemekh, 22(10):1345–1354, 1961.

**[Granmo (2018)]** Ole-Christoffer Granmo. The Tsetlin Machine - A Game Theoretic Bandit Driven Approach to
Optimal Pattern Recognition with Propositional Logic. arXiv preprint arXiv:1804.01508, 2018.

---

> ### Author Response · Authors · 2023-11-17
> **Summary of changes**
>
> We thank the reviewers again for dedicating their valuable time to our submission and we look forward to their further feedback.
>
> In response to the constructive feedback received, we have implemented several **key changes**:
>
> 1. We have revised the title and abstract to eliminate any potential confusion between Tsetlin Automaton and the Tsetline Machine model (page 1).
> 2. We extended the second paragraph of the introduction to emphasis the importance of the convergence, including some references (page 1).
> 3. We added a sentence at the end of the introduction to clarify that convergence proof for PCL, while important, does not imply convergence for TM (page 2).
> 4. We updated Section 2 to remove irrelevant details, enhancing overall clarity and focus (pages 2, 3).
> 5. We have included an algorithm detailing PCL in the appendix (page 11).
> 6. We have refined the description of Figure 2, offering additional details (page 3).
> 7. We added some experimental results in the appendix (page 11). Also, the code is available in our anonymous git link.
>
> We believe these modifications strengthen the overall quality of our submission.
> We welcome any further comments and appreciate your feedback.

---

### Author Response · Authors · 2023-11-22
**Gentle reminder**

Dear reviewers and AC,

We thank you again for your constructive comments.

As of November 14th, we submitted a rebuttal, addressing all your valuable feedback, and subsequently updated our manuscript.

With less than 24 hours remaining in ICLR's discussion period, we are eager to address any additional concerns you may have.

Thank you again for your time and consideration.


Best regards,


The authors of 5019.

---

### Meta-Review · Area_Chair_7ogd · 2023-12-15

**Metareview:**

This paper presents theoretical results on a novel variant of Tsetlin machines where a convergence proof can be given.  It has several weaknesses.  First, it does not settle any established open problem.  Second the paper is not accessible to those not already familiar with the area.  Third, the significance for the practice of machine learning is not made clear.

**Justification For Why Not Higher Score:**

See the top level comments.

**Justification For Why Not Lower Score:**

This is the lowest score.

---

### Decision · Program_Chairs · 2024-01-16

Reject